# Reducing antibiotic prescribing in primary care in England from 2014 to 2017: population-based cohort study

Xiaohui Sun,[iD] Martin C Gulliford[iD]

School of Population and Environmental Health Sciences, King's College London, London, UK

**Correspondence to**
Xiaohui Sun;
xiaohui.sun@kcl.ac.uk

## ABSTRACT

**Objective** To analyse individual-patient electronic health records to evaluate changes in antibiotic (AB) prescribing in England for different age groups, for male and female subjects, and by prescribing indications from 2014 to 2017.

**Methods** Data were analysed for 102 general practices in England that contributed data to the UK Clinical Practice Research Datalink (CPRD) from 2014 to 2017. Prescriptions for all ABs and for broad-spectrum β-lactam ABs were evaluated. Relative rate reductions (RRR) were estimated from a random-effects Poisson model, adjusting for age, gender, and general practice.

**Results** Total AB prescribing declined from 608 prescriptions per 1000 person-years in 2014 to 489 per 1000 person-years in 2017; RRR 6.9% (95% CI 6.6% to 7.1%) per year. Broad-spectrum β-lactam AB prescribing decreased from 221 per 1000 person-years in 2014 to 163 per 1000 person-years in 2017; RRR 9.3% (9.0% to 9.6%) per year. Declines in AB prescribing were similar for men and women but the rate of decline was lower over the age of 55 years than for younger patients. All AB prescribing declined by 9.8% (9.6% to 10.1%) per year for respiratory infections, 5.7% (5.2% to 6.2%) for genitourinary infections, but by 3.8% (3.1% to 4.5%) for no recorded indication. Overall, 38.8% of AB prescriptions were associated with codes that did not suggest specific clinical conditions, and 15.3% of AB prescriptions had no medical codes recorded.

**Conclusion** Antibiotic prescribing has reduced and become more selective but substantial unnecessary AB use may persist. Improving the quality of diagnostic coding for AB use will help to support antimicrobial stewardship efforts.

## INTRODUCTION

Antimicrobial resistance is a growing concern worldwide.[1 2] Many disease-causing pathogens have now developed resistance to antimicrobial drugs.[3] The pathways to high rates of antibiotic (AB) resistance at population level are complex but excessive AB use as medical care is often a proximal cause of AB resistance,[4 5] especially in communities.[6–8] Consequently, there are increasing calls for more carefully considered use of ABs in order to conserve the therapeutic potential of these drugs.[9] This

---

### Strengths and limitations of this study

► The study findings are derived from analysis of electronic health records data for more than 100 general practices in England that continuously contributed to the CPRD dataset over the 4-year study period.
► Comprehensive data for all antibiotic prescriptions and consultations at general practice surgeries were analysed.
► Antibiotic prescriptions issued outside general practices in out-of-hours settings were not captured.
► Antibiotic prescriptions may not always have been dispensed or taken by patients.

---

is particularly relevant in primary care, where more than 70% of all ABs are prescribed.[10 11] Inappropriate AB prescribing is known to be widespread in primary care.[12] Based on international comparisons, with both low[13] and high[14] AB prescribing being observed across Europe, without comparable variation in safety outcomes such as bacterial infections, it appears that a substantial reduction of present AB prescribing in primary care might be safe and feasible.

To deal with these concerns, aggregated data for AB prescribing are now being used for health service management. A contractual financial incentive, known as a 'quality premium', has been introduced into the English NHS for meeting indicative targets for year-on-year reductions in inappropriate AB use across all indications.[15] The English Surveillance Programme for Antimicrobial Utilisation and Resistance (ESPAUR)[10] analysed aggregated prescribing data and found that general practice AB prescriptions decreased by 13% between 2012 and 2016. Analysis of data for individual patients offers an opportunity for more detailed understanding of this decreasing trend. Dolk *et al*[16] analysed data from The Health Improvement Network (THIN) database from 2013 to 2015. They drew attention to limitations of primary care records as a data source,

BMJ

including the high proportion of AB prescriptions for which no 'clinical justification' was recorded.

The purpose of this study is to update data for AB prescribing trends in English general practices from 2014 to 2017. The analyses specifically aimed to provide estimates for the decline in AB use separately for male and female subjects and for people of different ages. We also aimed to evaluate which prescribing indications were most associated with reduced prescribing. We compared changes in all AB prescribing with changes in prescribing of broad-spectrum β-lactam ABs. Finally, we aimed to compare reductions in prescribing of individual major classes of ABs to provide complementary information.

## METHODS

### Data source

The UK Clinical Practice Research Datalink (CPRD)[17] was used as the data source for the study. This is a prospectively collected primary care database including data from approximately 7% of UK general practices. The total number of patients ever registered in CPRD is about 11 million, but the registered population has varied over time, and by 2017 there were approximately 2.5 million active UK patients. In the UK, more than 98% of the population are registered with a general practice and registrations are often maintained over many years. The CPRD is considered to be representative of the UK population.[17] Data collected in the CPRD are of high quality and include all medical diagnoses recorded at consultations and referrals, as well as all drug prescriptions issued by general practices.[18] For this study we included data from CPRD general practices in England, which participated in the data linkage scheme, and consistently contributed data in all years from 2014 to 2017. During this period the total number of general practices in the UK contributing to CPRD declined from 491 in 2014 to 285 in 2017. The number of CPRD general practices in England declined from 329 to 133, while the number participating in the data linkage scheme declined from 257 to 102 (online supplementary table 1). Individual participant data were included from 1 January 2014 or the start of the patient's CPRD record, whichever was the latest, to the 31 December 2017 or the end of the patient's CPRD record, whichever was the earliest. Data were obtained from the February 2018 release of CPRD. For practices that ended CPRD data collection during 2017, an equivalent end-of-year-date was also adopted for earlier years, because of the marked seasonality in AB use.

### Main measures

For each year of study, we calculated the person-time contributed by each patient between 1 January of the year, or start of registration if this was later, to 31 December of the year, or end of registration or date of death, if these were earlier. Person-time was employed as the denominator for rates. Prescriptions for ABs were identified using product codes for all AB drug classes included in

section 5.1 of the *British National Formulary* (BNF) except anti-tuberculous, anti-lepromatous agents, and methenamine, which were excluded.[19] The BNF groups ABs into the following categories: penicillins, cephalosporins (including carbapenems), tetracyclines, aminoglycosides, macrolides, clindamycin, sulfonamides (including combinations with trimethoprim), metronidazole and tinidazole, quinolones, drugs for urinary tract infection (nitrofurantoin), and other ABs.

We analysed broad-spectrum β-lactam ABs as a separate group, including the BNF category of 'broad-spectrum penicillins'[19] and cephalosporins. The category of 'broad-spectrum penicillins' includes ampicillin and amoxicillin and combinations with clavulanic acid or flucloxacillin. Carbapenems, which are only rarely used in primary care, were combined with cephalosporins for these analyses. Clinical indications for AB prescription were grouped into categories based on Read medical codes recorded into patients' clinical and referral records on the same date as the AB prescription, including 'respiratory conditions', 'genitourinary conditions', 'skin' conditions, 'eye' conditions, or no codes recorded (online supplementary tables 2 to 5). All other codes were grouped into a single category of 'other and non-specific codes'. The most frequently used codes in this category are shown in table 1 and included 'telephone encounter', 'patient reviewed', and 'telephone triage encounter'. Since specific coded indications for AB therapy were uncommon in this category, it is subsequently referred to as 'non-specific'. We analysed the prescription sequence variable to determine whether each prescription was the first in a sequence or whether it was a repeat prescription; the former were coded as 'acute' prescriptions and the latter were coded as 'repeat 'prescriptions.

### Statistical analysis

Antibiotic prescriptions for all ABs and broad-spectrum β-lactam ABs were enumerated by year. AB prescriptions of the same type on the same date were considered as a single event. Age was included as a continuous covariate but was also analysed in subgroups from 0 to 4 years, then 10-year age groups up to ≥85 years. Read codes recorded on the same date as an AB prescription were analysed according to indication. The primary indication on each date was allocated by giving priority to indications in the following sequence: respiratory, genitourinary, skin, and eye. We estimated AB prescription rates per 1000 person-years, and proportions of registered patients with ABs prescribed in a year in relation to age group, gender, study year, and main indication. In order to estimate annual changes in AB prescribing, we fitted it to hierarchical generalised linear Poisson models using the 'hglm' package[20] in the R programme. The dependent variable was a count of AB prescriptions (either all AB prescriptions or broad-spectrum β-lactam AB prescriptions). Predictors were calendar year, gender, and age, including quadratic and cubic terms to allow for non-linear effects of age. Calendar year was included as a linear predictor based on inspection of

**Table 1** Thirty most frequently used Read codes for 'other and non-specific' antibiotic prescribing indications.

| Read code | Read term | Number of events* |
|---|---|---|
| 9N31.00 | Telephone encounter | 51 504 |
| 6A…00 | Patient reviewed | 32 470 |
| 9N3A.00 | Telephone triage encounter | 26 900 |
| 246…00 | O/E - blood pressure reading | 25 502 |
| 242…00 | O/E - pulse rate | 15 918 |
| 9Z…00 | Administration NOS | 9278 |
| 22A…00 | O/E - weight | 8937 |
| 8CB…00 | Had a chat to patient | 8191 |
| 9N1C.11 | Home visit | 7813 |
| 1371 | Never smoked tobacco | 6065 |
| 9…00 | Administration | 5748 |
| 8CAL.00 | Smoking cessation advice | 5664 |
| 8B3H.00 | Medication requested | 5661 |
| 137S.00 | Ex-smoker | 4642 |
| 137P.00 | Cigarette smoker | 4565 |
| 9N3G.00 | SMS text message sent to patient | 3990 |
| 8B3S.00 | Medication review | 3891 |
| 8CA…00 | Patient given advice | 3838 |
| 246…11 | O/E - BP reading | 3810 |
| 9N4…00 | Failed encounter | 3514 |
| 661M.00 | Clinical management plan agreed | 3305 |
| 9N58.00 | Emergency appointment | 2930 |
| 1…00 | History/symptoms | 2827 |
| 212…00 | Patient examined | 2691 |
| 81H…00 | Dressing of wound | 2543 |
| 9Na…00 | Consultation | 2381 |
| 14L…00 | H/O: drug allergy | 2277 |
| 1969 | Abdominal pain | 2102 |
| 9N32.00 | Third-party encounter | 1948 |
| 679…11 | Advice to patient - subject | 1939 |

*Multiple codes per date were analysed.
BP, blood pressure; H/O, history of; NOS, not otherwise specified; O/E, on examination.

descriptive data and because non-linear effects would be difficult to estimate over a 4-year period. A random effect for general practice was included because of the repeated observations on general practices over years. The log of person-time was included as offset. Relative rate reductions were estimated as one minus the adjusted relative rate for the linear effect of calendar year. In view of the size of the dataset, we present confidence intervals rather than significance tests. Results were presented using the 'ggplot2' and 'forest plot' packages[21] in the R programme.[22]

## Research ethics
The research protocol for this study was submitted to and approved by the Medicines and Healthcare Products Regulatory Agency (MHRA) Independent Scientific Advisory Committee (ISAC), protocol 16_020. All patients' electronic health records analysed for this study were fully anonymised.

## Patient and public involvement
Neither patients nor public were involved in the development and design of this study, or the selection of outcome measures, or the conduct, analysis, and data dissemination of the study.

## RESULTS
### Overall antibiotic prescriptions
Analyses included 102 general practices that contributed data in each year from 2014 to 2017 (table 2). The registered population was 1.03 million in 2014 increasing to 1.07 in 2017. There were 539 219 AB prescriptions in 2014, declining to 459 476 in 2017. The AB prescribing rate declined from 608 per 1000 person-years in 2014 to 489 per 1000 person-years in 2017. The proportion of registered patients who were prescribed ABs in each year declined from just over 1 in 4 (25.3%) in 2014 to just over 1 in 5 (21.1%) in 2017. Figure 1 (left panel) shows changes in the proportion of patients prescribed ABs by year over the study period. A consistent year-on-year reduction was observed in each age-group from 0–4 years to ≥85 years. Marked AB prescribing variations were observed in relation to age, with the highest rates at the extremes of age.

A total of 195 750 broad-spectrum β-lactam AB prescriptions were made in 2014, declining to 153 423 in 2017. The proportion of all AB prescriptions that were broad-spectrum β-lactams decreased from 36.3% in 2014 to 33.4% in 2017 (table 2). Figure 1 (right panel) shows the change in proportion of patients prescribed broad-spectrum β-lactam ABs by age group. Although there was a year-on-year decrease in broad-spectrum β-lactam AB use in each age group, the absolute reduction appeared to be greater at older ages, where broad-spectrum β-lactam AB use was greatest.

Table 3 presents data for AB prescribing indications. Respiratory consultations accounted for the most frequent indication with 168 852 (31%) prescriptions in 2014 and 129 032 (28.1%) in 2017. Genitourinary infections and skin infections accounted for 9% and 7% of AB prescriptions, respectively, with little change over years. There were 77 431 (14%) AB prescriptions with no associated medical codes recorded in 2014 and 73 596 (16%) in 2017. There were 204 395 (38%) AB prescriptions with other and non-specific codes recorded in 2014 and 181 018 (39%) in 2017. Overall, more than half (54.1%) of the AB prescriptions were documented without specific clinical conditions recorded.

Table 4 shows the proportion of repeat prescriptions for different prescribing indications. In 2017, 78 166 (17%) AB prescriptions were recorded as repeat prescriptions. The proportion of repeat prescriptions was ≤2%

**Table 2** Numbers of antibiotic (AB) prescriptions, and AB prescribing rates, by year. Figures are frequencies except where indicated.

|  | 2014 | 2015 | 2016 | 2017 |
|---|---|---|---|---|
| General practices | 102 | 102 | 102 | 102 |
| Patients | 1 025 539 | 1 058 805 | 1 069 513 | 1 071 293 |
| Female (%) | 520 336 (50.7) | 536 082 (50.6) | 542 051 (50.7) | 543 324 (50.7) |
| Age (mean, SD, years) | 39.4 (23.4) | 39.5 (23.4) | 39.7 (23.5) | 39.9 (23.5) |
| Person-time (person-years) | 887 580 | 921 735 | 932 544 | 939 620 |
| All AB prescriptions | 539 219 | 494 185 | 482 917 | 459 476 |
| All AB prescribing rate (per 1000 person-years) | 608 | 536 | 518 | 489 |
| Proportion of patients prescribed AB (%) | 25.3 | 23.0 | 22.2 | 21.1 |
| Mean number of AB prescriptions in patients receiving prescriptions | 2.08 | 2.03 | 2.03 | 2.03 |
| Broad-spectrum β-lactam AB prescriptions | 195 750 | 174 353 | 167 056 | 153 423 |
| Broad-spectrum β-lactam AB prescribing rate (per 1000 person-years) | 221 | 189 | 179 | 163 |
| Proportion of patients prescribed broad-spectrum β-lactam AB (%) | 12.9 | 11.3 | 10.7 | 9.9 |
| Mean number of broad-spectrum β-lactam AB prescriptions in patients prescribed | 1.48 | 1.46 | 1.45 | 1.45 |

for respiratory, genitourinary, or eye conditions. For skin infections, 8% of AB prescriptions were recorded as repeat prescriptions. There were 10% of repeat prescriptions among AB prescribing associated with non-specific codes. Among 73 596 AB prescriptions in 2017 with no medical codes recorded, 56 216 (76%) were repeat prescriptions.

Informed by the apparent consistent annual declines in AB prescribing noted in table 2 and figure 1, figure 2 presents a Forest plot of annual relative reductions in AB prescribing adjusted for age, gender, and general practice. Estimates for all AB prescribing are shown in blue and for broad-spectrum β-lactam AB prescribing in red. The annual relative reduction in all AB prescribing was 6.9% (95% CI 6.6% to 7.1%). Estimates were generally similar for male and female subjects. For participants aged <55 years, the subgroup estimates were all greater

than the overall estimate, being greatest at age 45–54 years at 9.2% (8.4% to 9.9%) per year. For participants older than 55 years, estimates were consistently lower than the overall estimate being lowest at age 75–84 years and above at 4.3% (3.4% to 5.1%) per year. Considering subgroups of indications, rates of decline were greatest for respiratory indications (9.8%, 9.6% to 10.1%), and eye indications (11.0%, 9.9% to 12.2%). The rate of decline was smallest for AB prescriptions with no recorded indication (3.8%, 3.1% to 4.5%). The overall rate of decline was faster for broad-spectrum β-lactam ABs than all ABs at 9.3% (9.0% to 9.6%). Estimates were consistent for male and female subjects. The greatest relative decline was observed at 45–54 years (12.5%, 11.5% to 13.5%) and the lowest at 75–84 years (5.7%, 4.7% to 6.7%). The greatest decline was for skin condition indications (14.9%, 13.9% to 15.9%) and lowest for uncoded indications (5.5%, 4.5% to 6.4%).

### Changes in different classes of antibiotics

Figure 3 presents changes over time in the use of different classes of ABs. The most frequently issued ABs were penicillins, accounting for 56% of AB prescriptions in men and 44% in women in 2017; macrolides, men 14%, women 12%; tetracyclines, men 14%, women 12%; sulfonamide and trimethoprim combination, men 6%, women 11%. Clindamycin, aminoglycosides, and other ABs accounted for <1% of AB prescriptions and are not shown. During the period of study, drugs for urinary tract infections (nitrofurantoin) increased as a proportion of all AB prescriptions, in men from 2.6% in 2014 to 4.2% in 2017, and in women from 8.8% in 2014 to 13.7% in 2017. Tetracycline use also increased between 2014 and 2017, in men from 12.8% to 14.5% and in women from

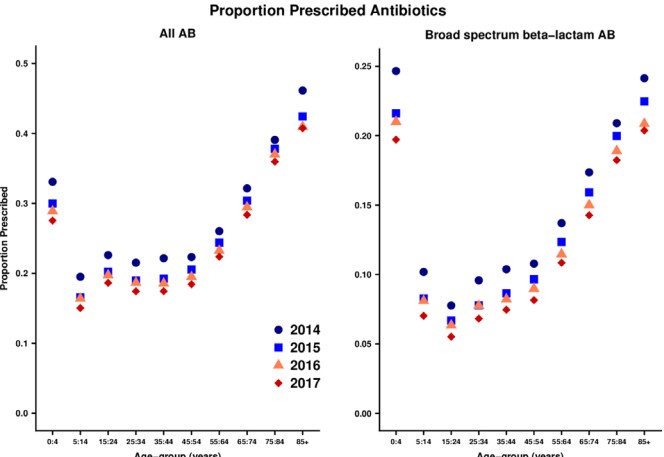

**Figure 1** Proportion of patients prescribed antibiotics (ABs) in one year by age-group and calendar year.

**Table 3** Distribution of antibiotic (AB) prescriptions by broad groups of indications. Figures are frequencies except where indicated

| | 2014 | | 2015 | | 2016 | | 2017 | | Total | |
|---|---|---|---|---|---|---|---|---|---|---|
| | Freq. | % | Freq. | % | Freq. | % | Freq. | % | Freq. | % |
| AB prescriptions | 539 219 | | 494 185 | | 482 917 | | 459 476 | | 1 975 797 | |
| Respiratory conditions | 168 852 | 31.3 | 146 025 | 29.5 | 140 263 | 29.0 | 129 032 | 28.1 | 584 172 | 29.6 |
| Genitourinary conditions | 47 009 | 8.7 | 44 544 | 9.0 | 42 453 | 8.8 | 42 401 | 9.2 | 176 407 | 8.9 |
| Skin conditions | 39 579 | 7.3 | 35 299 | 7.1 | 33 640 | 7.0 | 32 003 | 7.0 | 140 521 | 7.1 |
| Eye conditions | 1953 | 0.4 | 1622 | 0.3 | 1586 | 0.3 | 1426 | 0.3 | 6587 | 0.3 |
| Non-specific codes | 204 395 | 37.9 | 191 565 | 38.8 | 189 386 | 39.2 | 181 018 | 39.4 | 766 364 | 38.8 |
| No medical codes | 77 431 | 14.4 | 75 130 | 15.2 | 75 589 | 15.7 | 73 596 | 16.0 | 301 746 | 15.3 |

10.1% to 11.6%. Most other categories appeared to show slight declines. Both penicillin and macrolides were mainly prescribed for treating respiratory conditions, whereas tetracyclines was frequently issued for skin conditions among young patients and respiratory conditions in later life. There was a decline in the use of sulfonamide/trimethoprim combinations for urinary conditions while a notable increase of nitrofurantoin use for these conditions was seen over the study years among all age groups, but more particularly in women.

### Main findings

The rate of AB prescriptions and the proportion of patients receiving ABs have declined consistently over this 4-year period. Antibiotic use shows important variations by age and gender, being higher in very young and very old people and higher in women than men. However, these results show that a reduction in AB use is being achieved across all ages groups and in all subjects. The gender gap in relation to AB prescribing might be due to differences in medical care-seeking behaviour or specific conditions which disproportionally affect one gender.[23] Among prescriptions associated with coded indications, respiratory conditions were the most frequent indication for AB prescription and also showed the greatest rate of decline. Consistent with other recent reports,[16] we find that a substantial proportion of AB prescriptions are not associated with specific coded clinical indications and of these, a major share is associated with repeat prescriptions. Antibiotic prescriptions that were not associated

with medical codes showed the slowest rate of decline, potentially further identifying this category of prescriptions as representing a suboptimal standard of clinical practice which might hamper the accurate estimation of drug indications. Therefore, enhancing the quality of clinical information recording is warranted in order to improve patient care, and the usefulness of records for research and health service management.

More than one-third of prescriptions were for β-lactam ABs and there was evidence of an important decline in AB prescribing in this category consistent with previous evidence.[10] The relative reductions of broad-spectrum β-lactam prescriptions were greater than for overall AB use. Broad-spectrum β-lactam ABs may not necessarily offer more effective coverage of causal pathogens than their more specific counterparts. These results suggest that clinicians are gradually moving to more targeted narrow-spectrum substitutions when possible.

There is no universally accepted definition for 'broad-spectrum' ABs.[10 19] This study analysed a separate category of β-lactam ABs that were broad-spectrum (as 'broad-spectrum β-lactam ABs') to illustrate the possible difference in prescribing trends between these broad-spectrum ABs and their counterparts. For most common and uncomplicated infections, narrower spectrum drugs are generally recommended as first-line agents in general practices.[24] Macrolides are generally recommended as substitutions for penicillin in cases of penicillin allergy, and for specific indications, including Legionella or the

**Table 4** Proportion of antibiotic (AB) prescriptions that were either acute or repeat prescriptions in 2017. Figures are frequencies (percent of row total)

| | Total AB prescriptions | Acute | Repeat |
|---|---|---|---|
| AB prescriptions | 459 476 | 381 310 (83) | 78 166 (17) |
| Respiratory conditions | 129 032 | 127 474 (99) | 1558 (1) |
| Genitourinary conditions | 42 401 | 41 740 (98) | 661 (2) |
| Skin conditions | 32 003 | 29 513 (92) | 2490 (8) |
| Eye conditions | 1426 | 1399 (98) | 27 (2) |
| Non-specific codes | 181 018 | 163 804 (90) | 17 214 (10) |
| No medical codes | 73 596 | 17 380 (24) | 56 216 (76) |

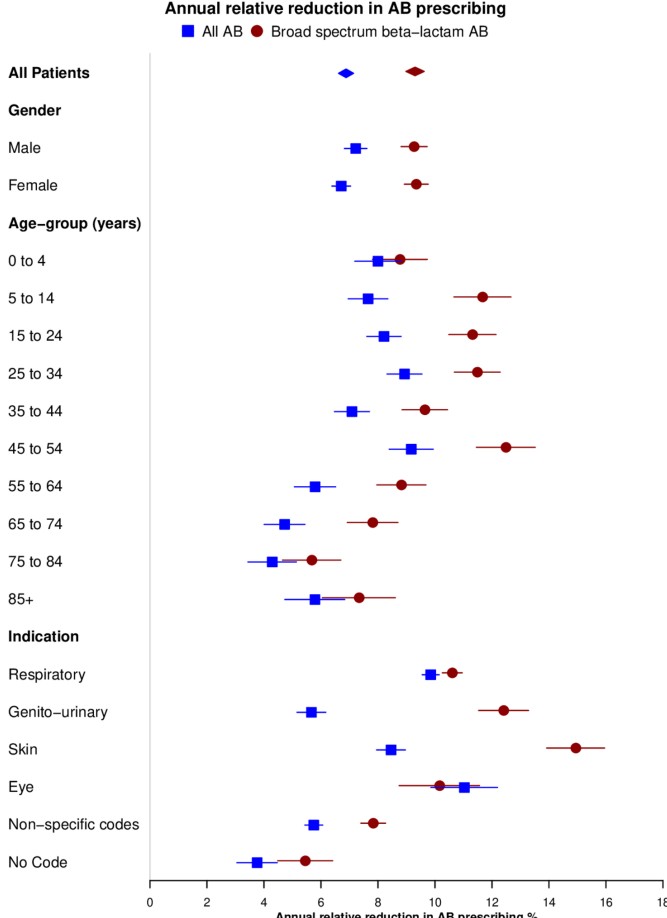

**Figure 2** Forest plot showing annual relative reduction (95% CI) in antibiotic (AB) prescribing for all ABs and broad-spectrum β-lactam ABs between 2014 and 2017 for subgroups of age and gender and different prescribing indications. Estimates were adjusted for age, gender, and clustering by practice.

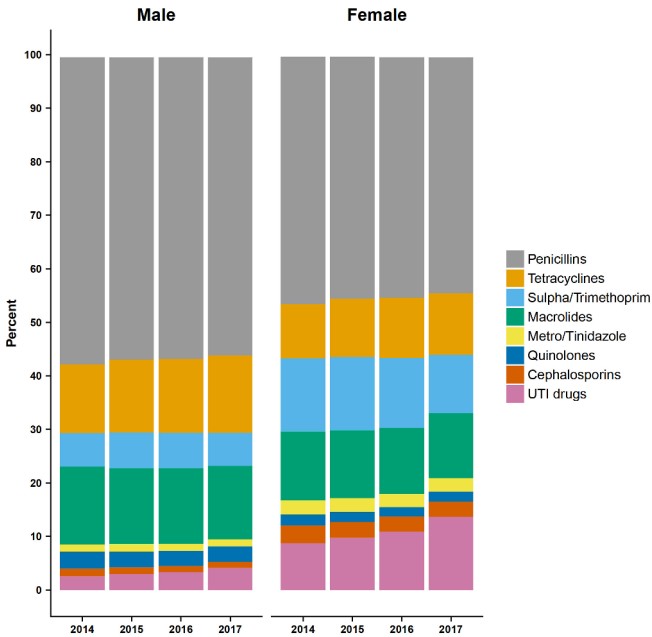

**Figure 3** Bar chart showing changes from 2014 to 2017 in the proportion of antibiotic prescriptions for different antibiotic classes for male and female subjects. UTI, urinary tract infection.

eradication of *Helicobacter pylori*. Nevertheless, macrolides were frequently prescribed in this and other studies.[25 26] Clinical use of tetracyclines was low in children in recognition of the risk of deposition in growing bone and teeth,[27] but the overall use of tetracyclines was higher at other ages. The increase of nitrofurantoin use was mainly due to the change in the guideline recommendation from trimethoprim to nitrofurantoin as empiric treatment for urinary tract infection.[24]

### Strengths and limitations

The study included more than 100 general practices in England that participated consistently across the 4-year period of study. The CPRD includes general practices from throughout the UK. However, because the CPRD licence imposes limits on the size of dataset to be employed, we selected only CPRD general practices in England. During the period of the study, there was substantial attrition of the cohort of CPRD general practices as practices migrated from the Vision practice systems that were employed by practices contributing to the CPRD database. We considered that it was important to include the same general practices in each year of study. However, we cannot be sure whether the AB prescribing of general practices that left the CPRD might differ from those that remained.

Previous studies have demonstrated the high quality and completeness of primary care electronic health records in CPRD.[17] The data suggested that repeat AB prescriptions might account for a high proportion of uncoded prescriptions, but the prescription sequence field has not been well-validated to our knowledge. A concern for this study is the possible lack of recording of out-of-hours prescriptions, especially those from deputising services, walk-in centres, and emergency care settings.[28] We noted that codes for telephone consultations and home visits were frequent among AB prescriptions with non-specific coded indications, which suggests that some out-of-hours activity might have been captured. We also acknowledge that prescriptions from hospitals and specialist clinics are not included, but these are expected to make only a small contribution to community AB use. It appears unlikely that the large and consistent reductions in prescribing seen in this paper could be accounted for by movement of prescribing to other care settings.

The research analysed prescriptions issued and not prescriptions dispensed or consumed by patients. We could not determine whether prescribers used a delayed or deferred AB prescribing strategy. For these reasons, we believe that AB consumption may be slightly lower than we have reported. We acknowledge that there are variations in prescribing between practices.[16 29 30] Our analytical method allowed us to estimate overall effects, and measures of precision, which took into account variation

between practices. Our results show some difference from an earlier study[16] in the distribution of indications, but since different general practices, from different databases, were included in the two studies this may reflect variations in clinical practice.

## Comparison with other studies

Previous analyses of primary care electronic health records have focused on AB prescribing for respiratory infections,[31 32] recognising that these conditions represent the most frequent indications for AB prescription. There has been a long-term decline in respiratory consultation rates in England, which has contributed to reducing AB use for these conditions.[31] Some authors suggest that respiratory consultations account for nearly two-thirds of AB use in primary care.[33] Our analyses are consistent with those of Dolk et al,[16] who found that respiratory consultations account for fewer than half of AB prescriptions. However, a high proportion of prescriptions may be associated either with no medical codes or non-specific codes, making interpretation difficult. There were further methodological differences between the study of Dolk et al[16] and our own. The former study relied on the THIN database with a different number of general practices participating in different years, and used code lists that might have differed in some respects. Consequently, minor numerical differences are to be expected.

## CONCLUSIONS

The present analyses add to recent reports by providing age- and gender-adjusted estimates of the rate of decline in AB use for all ABs and broad-spectrum β-lactam ABs, for different prescribing indications and different population subgroups defined by age and gender. The results show that the recent decline in AB use is broadly based and has occurred in all subgroups investigated. However, the decline in AB use has been at a faster rate for broad-spectrum β-lactam ABs than for all ABs; the decline is consistent by gender but tended to be lower over age 55 years; the slowest rate of decline is observed for AB prescriptions with no coded indications. The results emphasise the utility of electronic health records for providing individual-patient data for surveillance of trends in antimicrobial use and focusing future efforts at antimicrobial stewardship where these are most needed.

**Contributors** MCG and XS conceived the study. XS analysed and interpreted the data, MCG contributed additional analysis. XS wrote the draft of the manuscript and both authors revised and approved the final draft. XS is the guarantor.

**Funding** XS is supported by the China Scholarship Council. MCG was supported by the National Institute for Health Research (NIHR) Biomedical Research Centre at Guy's and St Thomas' NHS Foundation Trust and King's College London. This research is also supported by grants from the NIHR (HTA 13/88/10 and HS&DR 16/116/46).

**Competing interests** None declared.

**Patient consent for publication** Not required.

**Provenance and peer review** Not commissioned; externally peer reviewed.

**Data sharing statement** Clinical Practice ResearchDatalink data were analysed under licence and are not available for sharing.

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
