## [Reviewer comments · BMJ Open]

ARTICLE DETAILS

TITLE (PROVISIONAL)	Reducing antibiotic prescribing in primary care in England from 2014 to 2017. Population-based cohort study
AUTHORS	Sun, Xiaohui; Gulliford, Martin

VERSION 1 – REVIEW

REVIEWER	Nick Francis Cardiff University, Wales
REVIEW RETURNED	07-Jun-2018

GENERAL COMMENTS	I thought that this was a well-conducted and useful study which adds (although only a little) to our understand of the use of antibiotics in primary care. I would like to see the following points addressed: - In the methods you describe categorising the antibiotic prescriptions into a number of categories / types. I would like to see data on prescribing rates for each of these categories (this could a be supplementary table if space is short. I would also like to see these categories added to the forest plot. We would then be able to see how much quinolone prescribing, for example, has changed.- You do not seem to have taken into account repeat prescriptions. This can be determined from CPRD using the 'issue' field. This may partly explain the 'uncoded' prescriptions (i.e. Repeat prescriptions of tetracyclines for acne). I don't anticipate that there will be a large number of repeat prescriptions, but it would be very helpful to quantify this.- It would be useful to make it clearer that the prescriptions with 'other' codes are as unhelpful as those with no cards. In the discussion section you mention that codes for telephone consultations and home visits were frequent in the group with no diagnostic code, but this data is not presented in the results section - please add. Was this type of coding particularly prevalent in some practices?- The x axis in figure 1 should be labeled with the age category ranges.- You mention that some conditions disproportionately affect one gender - I presume this is primarily UTI in women. Please provide more detail.- There are some grammatical errors and stylistic issues. Suggest review to improve written English standard.
--

REVIEWER	Jeffrey A. Linder, MD, MPH, FACP Division of General Internal Medicine and Geriatrics, Northwestern University Feinberg School of Medicine, Chicago, IL USA
REVIEW RETURNED	12-Jul-2018

GENERAL COMMENTS	Summary Sun and Gulliford, in their manuscript “Reducing antibiotic prescribing in primary care in England from 2014 to 2017: population-based cohort study,” report data from the Clinical Practice Research Datalink (CPRD). The investigators identified all antibacterial antibiotics prescribed in 102 English General Practices and measured changes in antibiotic prescribing between 2014 and 2017 per person-time stratified by age, gender, indications, and whether the antibiotics were broader-spectrum. They found a 6.9% per year decrease in antibiotic prescribing, a 9.3% per year decline in broader-spectrum antibiotic prescribing, and noted a variety of differences by stratified variables. General Comments The topic is important and the data are interesting for those in antibiotic utilization research, but the data and discussion do not clearly point to concrete implications for ongoing efforts to reduce inappropriate antibiotic prescribing. The manuscript suffers from several major weaknesses. First, there are fundamental aspects of the analysis that the authors do not adequately define including the nature of the data source or the patient population. Does the data source capture all primary care prescribing? Specialty prescribing? The manuscript does not include a description of the patients (in, for example, a typical Table 1), practices, or the prescribers. “Enrollment” and “person time” are not defined. The reader can figure from Table 1 that person-time is 87% of the number of patients in each of the practices, but the authors do not describe how patients might move in, out, or between practices (or the data set). Is person-time assessed on a daily, monthly, or some other basis? It does not appear, as is customary, that there was some minimum enrollment criterion for a person to be included. (See Specific Comments for additional definitional concerns about the definitions of “antibiotics” and “indications.”) Second, the authors make the argument that the CPRD is representative of UK General Practice, but in the next sentence state that they limited the present analysis to practices in England that participated in a data linkage scheme and consistently contributed data from 2014 to 2017. Such restrictions would decrease the representativeness of the analysis. The CPRD has 11.3M patients, but only about 1M included in the present analysis. Considering generalizability beyond primary care practice, is the CPRD only representative of people who have or are engaged in primary care? Third, the authors are not clear about the purpose of their modeling technique. They state that the purpose of the analysis is to “evaluate changes in AB prescribing in England for different age-groups, for males and females, and by prescribing indications from 2014 to 2017,” but then go on to adjust for these same factors in their modeling. As such, it is unclear how all of the data provided in Figure 2 were calculated.
---

	Also in regards to modeling, the authors do not provide a rationale for the use of modeling linear changes in antibiotic prescribing over time. What if the changes are non-linear (see Mundkur, Drug Safety 2018)? The abstract stated that the models were adjusted for “general practice,” but this is not mentioned in the Methods. Was general practice modeled? Was it as a random or fixed effect? Fourth, the authors do not address the use of statistical testing. The authors do not state what they would consider a statistically significant or – if one did not want to use statistical testing because the sample size is so large – clinically significant difference. Also, the investigators could use interaction terms to evaluate differences between strata of gender, age, and indication rather than the qualitative terms in the manuscript. Fifth, the authors, in several places in the Abstract and Results, report results without providing actual data that should be present in the text, Tables, or Figures. As examples, the authors make statements about results, but provide no data on page 2, line 25; page 6, line 53 (and onto the next page); page 8, line 34; and page 9, line 2. Specific Comments General Comment: The manuscript contains grammatical errors, incorrect word use, and awkward phrasings throughout. General Comment: Throughout the manuscript, the authors use undefined abbreviations (starting with “AB” and “CPRD” in the Abstract). Page 2, Abstract Conclusion: This is simply a restatement of the results. What are the novel implications of this analysis? Page 2, line 52: The authors state that they did not capture “unclearly documented consultations,” but this appears to be a significant finding of theirs: 16% of antibiotic prescriptions had no medical codes recorded. Page 3, Line 8: The authors state that “reasons...[for] high rates of antibiotic resistance...are multifactorial,” but is it? It’s simply a result of bacterial exposure to antibiotics. Page 3, Line 22: This sentence is particularly awkward with a stray clause – “without risks to patient safety” – in the middle. Page 4, Line 49: It is unclear what the authors mean that “analysis time was included pro-rata for earlier years.” Page 4, Line 54: The authors should include more specificity about what antibiotics they included. In particular, it is concerning that they included aminoglycosides and carbapenems, which are exclusively intravenous. (It also does not instill confidence that the authors mention carbapenems on Page 5, Line 16 for inclusion in their definition of broader-spectrum antibiotics when not having mentioned carbapenems before.)
--	---

	Page 5, Line 16: The “indications” are not a cohesive variable. This variable conflates diagnoses (respiratory tract infections) and consultation/encounter types (out-of-hours consultations). Perhaps these are clear to a reader from the UK, but how are “out-of-hours consultations” an antibiotic indication? Are these encounters conducted by telephone, in-person, online portal, or something else? The codes mentioned on Page 7, line 39 are also not “indications.” In addition, the authors never examine what is contained in the “other codes recorded” category. Page 6, Line 45: This paragraph, I believe, contains a switch from analyses in person-time to per person. With such a switch, the authors need to address clustering of antibiotic prescriptions within individual patients (as one might infer from Table 1). Page 8, Line 42: The authors start using new, previously unmentioned metrics (antibiotics per person year, IRR). Page 22, Figure 1: This figure is not the most effective way to show changes in antibiotic prescribing over time, which appears to be the major point of the paper. Pages 19-23: In future submissions, the authors should be cognizant of how their manuscript appears to reviewers when converted to a PDF. Table 1 is cut-off on the right side. The use of color is not helpful on a black-and-white printed page (nor is the use of certain colors for the 8% of men who are colorblind). Figures 1 and 2 are very hard to read with the BMJ Open watermark and overlay in the header and footer.
--	---

VERSION 1 – AUTHOR RESPONSE

Reviewer: 1

I thought that this was a well-conducted and useful study which adds (although only a little) to our understand of the use of antibiotics in primary care.

Thank you for this feedback.

- In the methods you describe categorising the antibiotic prescriptions into a number of categories / types. I would like to see data on prescribing rates for each of these categories (this could a be supplementary table if space is short. I would also like to see these categories added to the forest plot. We would then be able to see how much quinolone prescribing, for example, has changed.

Thank you for this comment. We have now added a new Figure 3, which presents changes in the utilisation of different classes of antibiotics over time for males and females separately. We also discuss this graph in the main paragraph on page 9.

- You do not seem to have taken into account repeat prescriptions. This can be determined from CPRD using the 'issue' field. This may partly explain the 'uncoded' prescriptions (i.e. Repeat prescriptions of tetracyclines for acne). I don't anticipate that there will be a large number of repeat prescriptions, but it would be very helpful to quantify this.

Thank you for this comment. We now add (page 5): 'We used the 'issueseq' field in the CPRD therapy file to evaluate whether prescriptions might be repeat prescriptions in a series.'

We also report (page 8): 'Tabulation of the 'issueseq' field in the CPRD therapy file for 2017, suggested that 82.7% of prescriptions were first episodes, 5.7% were second episodes, 3.7% third episodes and 7.9% were fourth or higher.'

We also comment (page 12): 'The data suggested that repeat antibiotic prescriptions might be issued in a minority of cases but data in the 'issueseq' field has not been well-validated to our knowledge.'

- It would be useful to make it clearer that the prescriptions with 'other' codes are as unhelpful as those with no codes. In the discussion section you mention that codes for telephone consultations and home visits were frequent in the group with no diagnostic code, but this data is not presented in the results section - please add. Was this type of coding particularly prevalent in some practices?

Thank you, we now explain (page 8): 'There were 28% of prescriptions with 'other codes' recorded and 14% with 'out-of-hours' codes recorded. In 2017, the code for 'telephone encounter' accounted for 56% of 'out of hours codes', while codes for 'patient reviewed', 'had a chat to patient' and 'administration' accounted for 24% of 'other codes'. The proportion of antibiotic prescriptions with 'out-of-hours' codes ranged from 1% to 48% at different general practices, while the proportion with no codes recorded ranged from 8% to 29% at different practices.'

- The x axis in figure 1 should be labelled with the age category ranges.

Thank you Figure 1 has now been re-labelled as suggested.

- You mention that some conditions disproportionately affect one gender - I presume this is primarily UTI in women. Please provide more detail.

Thank you, we now comment (page 9): 'There was a decline in the use of sulphonamide/trimethoprim combinations for urinary conditions while a notable increase of nitrofurantoin use for these conditions was observed over study years among all age groups but more particularly in women.'

- There are some grammatical errors and stylistic issues. Suggest review to improve written English standard.

Thank you, we have carefully checked the revised manuscript and have corrected grammar and style.

Reviewer: 2

The topic is important and the data are interesting for those in antibiotic utilization research,

Thank you for this feedback.

First, there are fundamental aspects of the analysis that the authors do not adequately define including the nature of the data source or the patient population. Does the data source capture all primary care prescribing?

Thank you, we now explain (page 4): 'Data collected into CPRD are of high quality and include all medical diagnoses recorded at consultations and referrals, as well as all drug prescriptions issued by general practices. (18)'

Specialty prescribing?

Thank you, we now comment (page 12): 'We also acknowledge that prescriptions from hospitals and specialist clinics are not included, but these are expected to make only a small contribution to community antibiotic utilisation.'

The manuscript does not include a description of the patients (in, for example, a typical Table 1), practices, or the prescribers.

Thank you, we now explain (page 4): 'In the UK, more than 98% of the population are registered with a general practice and registrations are often maintained over many years.' We also add information about age and gender distribution of the samples to the revised Table 1.

"Enrollment" and "person time" are not defined. The reader can figure from Table 1 that person-time is 87% of the number of patients in each of the practices, but the authors do not describe how patients might move in, out, or between practices (or the data set). Is person-time assessed on a daily, monthly, or some other basis? It does not appear, as is customary, that there was some minimum enrollment criterion for a person to be included. (See Specific Comments for additional definitional concerns about the definitions of "antibiotics" and "indications.")

Thank you, we now explain (page 4): 'For each year of study, we calculated the person-time contributed by each patient between 1st January of the year, or start of registration if this was later, to 31st December of the year, or end of registration or date of death, if these were earlier.'

Second, the authors make the argument that the CPRD is representative of UK General Practice, but in the next sentence state that they limited the present analysis to practices in England that participated in a data linkage scheme and consistently contributed data from 2014 to 2017.

Thank you for these comments, we now explain (page 4): 'Antibiotic prescriptions are frequent events and could lead to an excessively large dataset. Therefore, for this study we included data from CPRD general practices in England, which participated in the data linkage scheme, and consistently contributed data from 2014 to 2017.'

Such restrictions would decrease the representativeness of the analysis. The CPRD has 11.3M patients, but only about 1M included in the present analysis. Considering generalizability beyond primary care practice, is the CPRD only representative of people who have or are engaged in primary care?

Thank you, we now explain (page 4): 'The total number of patients ever registered in CPRD is about 11 million, but the registered population has varied over time and by 2017 there were approximately 2.5 million active UK patients.'

Third, the authors are not clear about the purpose of their modelling technique. They state that the purpose of the analysis is to "evaluate changes in AB prescribing in England for different age-groups, for males and females, and by prescribing indications from 2014 to 2017," but then go on to adjust for these same factors in their modelling. As such, it is unclear how all of the data provided in Figure 2 were calculated.

Thank you for this comment. We now explain more clearly (page 6): 'In order to estimate annual changes in antibiotic prescribing, we fitted in hierarchical generalized linear Poisson models using the 'hglm' package (20) in the R program. The dependent variable was a count of antibiotic prescriptions (either all AB prescriptions or BS-AB prescriptions). Predictors were calendar year, gender and age, including quadratic and cubic terms to allow for non-linear effects of age. Calendar year was included as a linear predictor because non-linear effects would be difficult to estimate over four years. A

random effect for general practice was included because of the repeated observations on general practices over years. The log of person-time was included as offset. Relative rate reductions were estimated as one minus the adjusted relative rate for the linear effect of calendar year.'

Also in regards to modeling, the authors do not provide a rationale for the use of modeling linear changes in antibiotic prescribing over time. What if the changes are non-linear (see Mundkur, Drug Safety 2018)?

Thank you, we now add on page 6: 'Calendar year was included as a linear predictor because non-linear effects would be difficult to estimate over four years.'

The abstract stated that the models were adjusted for "general practice," but this is not mentioned in the Methods. Was general practice modeled? Was it as a random or fixed effect?

Thank you, we now explain on page 6: 'A random effect for general practice was included because of the repeated observations on general practices over years.'

Fourth, the authors do not address the use of statistical testing. The authors do not state what they would consider a statistically significant or – if one did not want to use statistical testing because the sample size is so large – clinically significant difference.

Thank you, we now explain (page 6): 'In view of the size of the dataset, we present confidence intervals rather than significance tests.'

Also, the investigators could use interaction terms to evaluate differences between strata of gender, age, and indication rather than the qualitative terms in the manuscript.

Thank you for this comment, as noted above, hypothesis testing can be problematic in a large dataset because small differences may be 'statistically significant' and because of the potential multiplicity of hypotheses. This epidemiological study was mainly interested in presenting descriptive results of antibiotic prescription trends over time and providing implications for further studies.

Fifth, the authors, in several places in the Abstract and Results, report results without providing actual data that should be present in the text, Tables, or Figures. As examples, the authors make statements about results, but provide no data on page 2, line 25; page 6, line 53 (and onto the next page); page 8, line 34; and page 9, line 2.

Thank you, we have adjusted the wording to ensure all interpretations are justified.

General Comment: The manuscript contains grammatical errors, incorrect word use, and awkward phrasings throughout.

Thank you, we have now comprehensively revised the manuscript as noted above.

General Comment: Throughout the manuscript, the authors use undefined abbreviations (starting with "AB" and "CPRD" in the Abstract).

Thank you, all abbreviations have now been explained.

Page 2, Abstract Conclusion: This is simply a restatement of the results. What are the novel implications of this analysis?

Thank you, we now comment (page 2): ‘Antibiotic prescribing has reduced and become more selective but substantial unnecessary AB utilisation may persist. Improving the quality of coded indications for AB utilisation will help to support antimicrobial stewardship efforts.’

Page 2, line 52: The authors state that they did not capture “unclearly documented consultations,” but this appears to be a significant finding of theirs: 16% of antibiotic prescriptions had no medical codes recorded.

Apologies, we are not able to locate this comment on page 2?

Page 3, Line 8: The authors state that “reasons...[for] high rates of antibiotic resistance...are multifactorial,” but is it? It’s simply a result of bacterial exposure to antibiotics.

Thank you, we have adjusted the wording so that it now reads (page 3): ‘The pathways to high rates of antibiotic resistance at population level are complex but excessive medical utilisation is often a proximal cause of antibiotic resistance (4, 5) especially in community settings.(6-8)’

Page 3, Line 22: This sentence is particularly awkward with a stray clause – “without risks to patient safety” – in the middle.

Thank you, we now say (page 3): ‘Based on international comparisons, with both low- (13) and high- (14) antibiotic prescribing being observed across Europe, without comparable variation in safety outcomes such as bacterial infections,..’

Page 4, Line 49: It is unclear what the authors mean that “analysis time was included pro-rata for earlier years.”

Thank you, we now explain (page 4): ‘For practices that ended CPRD data collection during 2017, an equivalent end-of-year-date was also adopted for earlier years, because of the marked seasonality in antibiotic utilisation.’

Page 4, Line 54: The authors should include more specificity about what antibiotics they included. In particular, it is concerning that they included aminoglycosides and carbapenems, which are exclusively intravenous. (It also does not instill confidence that the authors mention carbapenems on Page 5, Line 16 for inclusion in their definition of broader-spectrum antibiotics when not having mentioned carbapenems before.)

Thank you, we now explain (page 5) that we included: ‘all antibiotic drug classes included in section 5.1 of the British National Formulary (BNF) except anti-tuberculous, anti-lepromatous agents and methenamine, which were excluded (20).’

We also explain (page 5) that ‘The BNF groups antibiotic drugs into the following categories: penicillins, cephalosporins (including carbapenems)... ‘

Page 5, Line 16: The “indications” are not a cohesive variable. This variable conflates diagnoses (respiratory tract infections) and consultation/encounter types (out-of-hours consultations). Perhaps these are clear to a reader from the UK, but how are “out-of-hours consultations” an antibiotic indication? Are these encounters conducted by telephone, in-person, online portal, or something else? The codes mentioned on Page 7, line 39 are also not “indications.” In addition, the authors never examine what is contained in the “other codes recorded” category.

Thank you for this comment. We now refer (page 4) to 'prescribing indications, and other coded reasons for prescription'

Page 6, Line 45: This paragraph, I believe, contains a switch from analyses in person-time to per person. With such a switch, the authors need to address clustering of antibiotic prescriptions within individual patients (as one might infer from Table 1).

Page 8, Line 42: The authors start using new, previously unmentioned metrics (antibiotics per person year, IRR).

Page 22, Figure 1: This figure is not the most effective way to show changes in antibiotic prescribing over time, which appears to be the major point of the paper.

Thank you, our aim in this figure is not only to show data for changes over time, but also to illustrate the important variations in AB utilisation by age-group and gender. With respect, we consider that age and gender related differences are worthy of illustration in their own right. Our main estimates for changes over time are presented in Figure 2.

Pages 19-23: In future submissions, the authors should be cognizant of how their manuscript appears to reviewers when converted to a PDF. Table 1 is cut-off on the right side. The use of color is not helpful on a black-and-white printed page (nor is the use of certain colors for the 8% of men who are colorblind). Figures 1 and 2 are very hard to read with the BMJ Open watermark and overlay in the header and footer.

Thank you for this feedback. Table 1 is shown in landscape format and we are not sure why this has not carried over into the pdf. We have adjusted our selection of colours using a colour-blind friendly palette for this plot, and have also used different plotting symbols in the Figures.

VERSION 2 – REVIEW

REVIEWER	Nick Francis Cardiff University, UK
REVIEW RETURNED	18-Oct-2018

GENERAL COMMENTS	Thank you for sending this revised manuscript, which I think is an improvement on the previous version. I think the addition of antibiotic class (figure 3) adds significantly to the paper. I previously suggested that you use the issueseq field in the therapy table to identify repeat prescribing. From what you have written it is unclear to me whether you understand how prescriptions are issued and coded in UK EMRs, and I am confused by what you have written. Prescriptions can be issued as acute or repeat prescriptions. Repeat prescriptions are added they include a number indicating how many times they can be issued before requiring reauthorisation. Prescriptions can then be issued without a consultation, with each issue numbered (e.g. from 1 to 6) up until the maximum authorised number has been reached. Acute prescriptions do not have an issue number, and therefore are coded as '0' in the issueseq field in CPRD. This is important in relation to antibiotic prescribing because antibiotic prescriptions for acute infections would be expected to be issued as acute
--

prescriptions and antibiotics issued as repeat prescriptions would be expected to be issued for other reasons (chronic conditions like acne, prophylaxis, rescue packs, etc.). The relevant question is whether the prescription is acute (issueseq=0) or repeat (issueseq>0). The number recorded in issueseq is not relevant in an analysis like this. With this in mind, I'm not sure what exactly is meant by your statement, '.. 82.7% of prescriptions were first episodes, 5.7% were second episodes, 3.7% third episodes and 7.9% fourth or higher'. Rather than tabulating antibiotic prescriptions by issue number, what would be helpful would be classifying prescriptions as acute (issueseq=0) or repeat (issueseq>0) and using this data to help explain the data. Giving the crude percentage that are acute vs repeat is a start, but this information can also be used to try and explain prescriptions without indications (an indication for repeat prescribing is only likely to be added when the prescription was first added, and therefore it would not be unusual for there to be many issues that did not have any coded 'indication') and to describe changes by age and over time by whether they are acute or repeat.

I'm still confused about how you analysed the coded clinical and consultation data. Any infection diagnoses will be coded in medcode field in the clinical table. As you indicate, and has been documented previously, many antibiotic prescriptions (even if you limit to acute prescriptions) occur in consultations where there is no code indicating a relevant infection diagnosis. Sometimes there are codes for symptoms or other relevant codes, but some codes such as 'telephone consultation' or 'patient reviewed' give no indication what the indication for the prescription was. There are also codes used in the consultation table that indicate the type of consultation. You have indicated that 28% of prescriptions had 'other codes' and 14% had 'out of hours codes'. I presume that when you refer to 'out of hours' codes you are referring to consultations that have a medcode indicating that it was an out of hours consultation rather than it being coded as an out of hours consultation in the consultation type field. If so, why are you presenting this as a separate category from other? It is very difficult to interpret the meaning of medcodes that do not indicate a specific diagnosis. Sometimes the mecode indicates a consultation type that differs from the consultation type recorded in the consultation table. Unless you can provide a clear justification for presenting 'out of hours' codes as a separate category I suggest including them in the 'other codes' category and then describing the most frequent codes that make up the 'other codes' category. At the moment you describe 'patient reviewed', 'had a chat to patient' and 'administration' as making up 24% of the other codes, and 'telephone encounter' (which is a type of consultation) as making up 56% of the 'out of hours' codes. I don't understand how 'telephone encounter' can make up 56% of the 'out of hours' coded consultations unless 56% of the consultations that had an 'out of hours' medcode also had a 'telephone encounter' medcode, or you are using the consultation type field in the consultation table. Please explain? Finally, as I mentioned previously, the helpful distinction is consultations that include one or more codes that indicate a diagnosis (indication) and those that don't. To that end, consultations with only a vague code (such as 'patient reviewed') are as unhelpful as those with no codes. I would suggest tabulating consultations by diagnostic code (RTI, GU, Skin, Eye, no indication given (which would include non-specific codes and no codes)) and type of prescription (acute, repeat).

	Please include the code lists used as supplementary files.
--	--

REVIEWER	Linder, Jeffrey Northwestern University Feinberg School of Medicine
REVIEW RETURNED	11-Nov-2018

GENERAL COMMENTS	Summary Thank you for the opportunity to re-review the article by Sun and Gulliford, in their manuscript “Reducing antibiotic prescribing in primary care in England from 2014 to 2017: population-based cohort study.” I have modified my original comments below. The investigators identified all antibacterial antibiotics prescribed in 102 English General Practices and measured changes in antibiotic prescribing between 2014 and 2017 per person-time stratified by age, gender, indications, and whether the antibiotics were broader-spectrum. They found a 6.9% per year decrease in antibiotic prescribing, a 9.3% per year decline in broader-spectrum antibiotic prescribing, and noted a variety of differences by stratified variables. General Comments The topic is important and contains good and bad news. The good news is that there was a decrease in antibiotic prescribing in General Practice in England during the study. The decrease appears driven by decreases in the proportion of patients prescribed an antibiotic (or a broad-spectrum antibiotic) and by decreases in respiratory infection prescribing. The bad news is there remains a lot of antibiotic prescribing – potentially over half – that is at best poorly documented and at worst poorly indicated (other codes, out-of-hours prescribing, and no medical codes). As I wrote before, the data are very interesting to those of us in antibiotic utilization research. A major improvement is that the discussion more clearly points to concrete implications for ongoing efforts to reduce inappropriate antibiotic prescribing which will raise the interest among a broader audience. In the revision, the authors have done a nice job addressing my prior comments about defining aspects of the analysis (e.g., person-time, degree of prescribing captured), the structure of their multivariable models, the use (or non-use) of statistical testing, grammatical problems, indication coding, and implications of their findings. A few moderate problems persist. First, the authors still do not make a convincing argument for the representativeness of the included practices or describe how the results might differ for the English population not captured in the present analysis. The response about the “excessively large data set” is a data handling challenge and is not an answer to the potential representativeness problem (it is probably the cause). The response about the present dataset having 1M of the 2.5M active UK patients also does not address representativeness. The
--

	authors need to provide a rationale or data showing that the data they present are representative. Second, the authors still do not provide an explicit, adequate rationale for linear modeling of antibiotic prescribing over time. However, the answer lies in Table 1. The raw data shows a near linear decrease in the antibiotic prescribing rate making it reasonable to model antibiotic prescribing over time linearly. Third, the authors have introduced a technical response that means little to the reader. They mention the “issueseq” field in the Methods, Results, and Discussion, but do not define it. The authors should define it in the Methods then use the actual, underlying concept throughout the rest of the manuscript. Fourth, it remains concerning that the manuscript continues to mention carbapenems so prominently. I realize my prior comment about mention of cabapenems “not instill[ing] confidence” was vague. To clarify, carbapenems – imipenem, ertapenem, meropenem, and doripenem in the US – are intravenous antibiotics reserved for use in multi-drug resistant bacterial infections, generally in extremely sick, hospitalized patients. Carbapenems are not used in primary care. Mention of carbapenems in this article gives the impression that the investigators do not know the subject. If carbapenems are found in the data in anything but the smallest of numbers, I would be very concerned about the quality of the data. As written, carbapenems are mentioned in the Abstract, Methods, and the footnote of Table 1. The authors should limit mention of carbapenems to the Methods in which they define antibiotics and broad-spectrum antibiotics. Related -- and apologies for being harsh -- the paragraph on the definition of antibiotics and broad-spectrum antibiotics is a bit of a mess. The manuscript only defines “broad-spectrum beta-lactam antibiotics” and does not say what the investigators considered non-beta-lactam broad-spectrum antibiotics. Carbapenems are considered a subset of cephalosporins (page 5, line19), but are also considered their own broad-spectrum beta-lactam antibiotic class. The authors need to rewrite this whole section and state explicitly what they considered “antibiotics” and “broad-spectrum antibiotics.” They can simply cite or explicitly refer to BNF and ESPAUR. Specific Comments Page 7, line 41: This sentence makes it sound as if there is gender-specific data in Figure 1, but there is not. Page 11, line 21: This sentence about macrolides is hard to follow. Do the authors mean that the “existing strong indications” are rare and would not explain the amount of macrolide prescribing seen?
--	---

VERSION 2 – AUTHOR RESPONSE

Response to comments:

Reviewer: 1

-Thank you for sending this revised manuscript, which I think is an improvement on the previous version. I think the addition of antibiotic class (figure 3) adds significantly to the paper.

Thank you for this feedback.

I previously suggested that you use the `issueseq` field in the therapy table to identify repeat prescribing.... Rather than tabulating antibiotic prescriptions by issue number, what would be helpful would be classifying prescriptions as acute (`issueseq=0`) or repeat (`issueseq>0`) and using this data to help explain the data. Giving the crude percentage that are acute vs repeat is a start, but this information can also be used to try and explain prescriptions without indications.

Thank you for this helpful explanation and advice. We have now used the `'issueseq'` field to classify antibiotic prescriptions as `'acute'` (`issueseq==0`) or `'repeat'` (`issueseq>0`). We provide a new Table 3, which gives a tabulation of the proportion of repeat prescriptions by prescription indication. The Table shows that only a small minority of antibiotic prescriptions for clearly diagnosed indications were repeat prescriptions (except for skin infections), while prescriptions with non-specific codes or with no codes recorded were more likely to be repeat prescriptions, as anticipated by the Reviewer.

We now explain in the Methods section (Pages 5-6): `'We used the 'issueseq' field in the CPRD therapy file to evaluate whether prescriptions were repeat prescriptions. Prescriptions associated with 'issueseq' values of zero were coded as 'acute' while 'issueseq' values of one or above were coded as 'repeat' prescriptions.'`

We also comment in the Results section (pages 8-9): `'Table 3 shows the proportion of repeat prescriptions for different prescribing indications. In 2017, 78,166 (17%) of antibiotic prescriptions were recorded as repeat prescriptions. The proportion of repeat prescriptions was 2% or lower for respiratory, genitourinary or eye conditions. For skin infections, 8% of antibiotic prescriptions were recorded as repeat prescriptions. There were 10% of repeat prescriptions among antibiotic prescribing episodes associated with non-specific codes. Among 73,596 antibiotic prescriptions in 2017 with no medical codes recorded, 56,216 (76%) were recorded as repeat prescriptions.'`

We also note in the Discussion (Page 11): `'These results suggested that prescriptions without coded indications included a high proportion of repeat prescriptions.'`

I'm still confused about how you analysed the coded clinical and consultation data. ...Unless you can provide a clear justification for presenting `'out of hours'` codes as a separate category I suggest

including them in the 'other codes' category and then describing the most frequent codes that make up the 'other codes' category.

Thank you for this feedback. We have now combined the categories of 'Other Codes' and 'Out of Hours' codes into the category of 'Non-specific codes'. This category is now shown in Tables 2 and 3 as well as Figure 2.

Finally, as I mentioned previously, the helpful distinction is consultations that include one or more codes that indicate a diagnosis (indication) and those that don't. To that end, consultations with only a vague code (such as 'patient reviewed') are as unhelpful as those with no codes. I would suggest tabulating consultations by diagnostic code (RTI, GU, Skin, Eye, no indication given (which would include non-specific codes and no codes)) and type of prescription (acute, repeat).

Thank you, a new Table 3 now presents a tabulation of the proportion of 'acute' and 'repeat' prescriptions according to main prescribing indications, as discussed above.

Please include the code lists used as supplementary files.

Thank you, our code lists are now presented as a supplementary document as suggested. We refer to this on page 5 'Codes employed for analysis are shown in Supplementary Tables 2 to 5.' We also add (pages 5-6) 'We adopted an inclusive approach to code selection in order to capture any potential indications for antibiotic treatment.' We checked and revised the code lists for this revision and this accounts for slightly different numbers from those reported previously.

Reviewer: 2

As I wrote before, the data are very interesting to those of us in antibiotic utilization research. A major improvement is that the discussion more clearly points to concrete implications for ongoing efforts to reduce inappropriate antibiotic prescribing which will raise the interest among a broader audience. In the revision, the authors have done a nice job addressing my prior comments about defining aspects of the analysis (e.g., person-time, degree of prescribing captured), the structure of their multivariable models, the use (or non-use) of statistical testing, grammatical problems, indication coding, and implications of their findings.

Thank you for this feedback.

First, the authors still do not make a convincing argument for the representativeness of the included practices or describe how the results might differ for the English population not captured in the present analysis. The response about the "excessively large data set" is a data handling challenge and is not an answer to the potential representativeness problem (it is probably the cause). The response about the present dataset having 1M of the 2.5M active UK patients also does not address

representativeness. The authors need to provide a rationale or data showing that the data they present are representative.

Thank you for this comment. In order to increase the transparency of general practice selection, we now provide a Supplementary Table 1 that shows the number of general practices contributing to CPRD over time. We now add further explanation on page 4: 'For this study we included data from CPRD general practices in England, which participated in the data linkage scheme, and consistently contributed data in all years from 2014 to 2017. During this period the total number of general practices in the UK contributing to CPRD declined from 491 in 2014 to 285 in 2017. The number of CPRD general practices in England declined from 329 to 133, while the number participating in the data linkage scheme declined from 257 to 102. (Supplementary Table 1).'

We also comment on page 13: 'The CPRD includes general practices from throughout the UK. However, because the CPRD licence imposes limits on the size of dataset to be employed, we selected only CPRD general practices in England. During the period of the study, there was substantial attrition of the cohort of CPRD general practices as practices migrated from the Vision practice systems that was employed by practices contributing to the CPRD database. We considered that it was important to include the same general practices in each year of study, with more than 100 general practices included in total. However, we cannot be sure whether the antibiotic prescribing of general practices that left the CPRD might differ from those that remained.'

Second, the authors still do not provide an explicit, adequate rationale for linear modeling of antibiotic prescribing over time. However, the answer lies in Table 1. The raw data shows a near linear decrease in the antibiotic prescribing rate making it reasonable to model antibiotic prescribing over time linearly.

Thank you for this comment. We now add on page 9: 'Informed by the apparent consistent annual declines in antibiotic prescribing noted in Table 1 and Figure 1, Figure 2 presents a Forest plot of annual relative reductions in AB prescribing adjusted for age, gender and general practice.'

Third, the authors have introduced a technical response that means little to the reader. They mention the "issueseq" field in the Methods, Results, and Discussion, but do not define it.

Thank you, please see our response to Reviewer 1.

Fourth, it remains concerning that the manuscript continues to mention carbapenems so prominently. I realize my prior comment about mention of carbapenems "not instill[ing] confidence" was vague. To clarify, carbapenems – imipenem, ertapenem, meropenem, and doripenem in the US – are intravenous antibiotics reserved for use in multi-drug resistant bacterial infections, generally in extremely sick, hospitalized patients. Carbapenems are not used in primary care. Mention of carbapenems in this article gives the impression that the investigators do not know the subject. If carbapenems are found in the data in anything but the smallest of numbers, I would be very

concerned about the quality of the data. As written, carbapenems are mentioned in the Abstract, Methods, and the footnote of Table 1. The authors should limit mention of carbapenems to the Methods in which they define antibiotics and broad-spectrum antibiotics.

Related -- and apologies for being harsh -- the paragraph on the definition of antibiotics and broad-spectrum antibiotics is a bit of a mess. The manuscript only defines "broad-spectrum beta-lactam antibiotics" and does not say what the investigators considered non-beta-lactam broad-spectrum antibiotics. Carbapenems are considered a subset of cephalosporins (page 5, line19), but are also considered their own broad-spectrum beta-lactam antibiotic class. The authors need to rewrite this whole section and state explicitly what they considered "antibiotics" and "broad-spectrum antibiotics." They can simply cite or explicitly refer to BNF and ESPAUR.

Thank you, we now explain (page 5): 'There is no universally accepted definition for 'broad-spectrum' antibiotics. (10, 19) For the present study, broad-spectrum β -lactam antibiotics included broad-spectrum penicillins and cephalosporins. Carbapenems are only rarely used in primary care and were combined with cephalosporins for further analysis.'

We also add (page 12): 'There is no universally agreed definition for 'broad-spectrum' antibiotics, therefore this study employed the category of β -lactam antibiotics that were broad-spectrum (as 'broad-spectrum beta-lactam antibiotics') to illustrate the possible difference in prescribing trends between these broad-spectrum antibiotics and their counterparts.'

We have now omitted mention of carbapenems from the Abstract and we have omitted the footnote to Table 1.

Specific Comments

Page 7, line 41: This sentence makes it sound as if there is gender-specific data in Figure 1, but there is not.

Thank you for this comment. These sentences referred to additional analysis but as this is not shown, the remarks about gender differences have now been omitted.

Page 11, line 21: This sentence about macrolides is hard to follow. Do the authors mean that the "existing strong indications" are rare and would not explain the amount of macrolide prescribing seen?

Thank you for this comment, we now explain (page 12): 'Macrolides are generally recommended as substitutions for penicillin in the case of penicillin allergy, as well as for specific indications including

Legionella or the eradication of Helicobacter pylori (HP). Nevertheless, macrolides were frequently prescribed in this and other studies. (25, 26).’

VERSION 3 – REVIEW

REVIEWER	Nick Francis Cardiff University
REVIEW RETURNED	01-Feb-2019

GENERAL COMMENTS	I was very pleased to review this paper again. All of my concerns have now been addressed and I feel that this is a very interesting paper.
---

REVIEWER	Jeffrey A. Linder, MD, MPH, FACP Northwestern University Feinberg School of Medicine
REVIEW RETURNED	22-Feb-2019

GENERAL COMMENTS	Summary Thank you for the opportunity to review the second revision of the article by Sun and Gulliford, in their manuscript “Reducing antibiotic prescribing in primary care in England from 2014 to 2017: population-based cohort study.” I have modified my original comments below. The investigators identified all antibacterial antibiotics prescribed in 102 English General Practices and measured changes in antibiotic prescribing between 2014 and 2017 per person-time stratified by age, gender, and indications, and whether the antibiotics were broader-spectrum beta-lactam antibiotics. They found a 6.9% per year decrease in antibiotic prescribing, a 9.3% per year decline in broader-spectrum beta-lactam antibiotic prescribing, and noted a variety of differences by stratified variables. General Comments The topic is important and contains good and bad news. The good news is that there was a decrease in antibiotic prescribing in General Practice in England during the study. The decrease appears driven by decreases in the proportion of patients prescribed an antibiotic (or a broad-spectrum antibiotic) and by decreases in respiratory infection prescribing. The bad news is there remains a lot of antibiotic prescribing – potentially over half – that is at best poorly documented and at worst poorly indicated (other codes, out-of-hours prescribing, and no medical codes). As I wrote before, the data are very interesting to those of us in antibiotic utilization research. From the initial submission to the first revision the investigators improved the discussion, aspects of the analysis (e.g., person-time, degree of prescribing captured), the structure of their multivariable models, the use (or non-use) of
--

	statistical testing, grammatical problems, indication coding, and implications of their findings. From the first revision to the present version the authors have addressed the representativeness of the data, provided a rationale for linear modeling of time, and addressed that they only looked at broader-spectrum beta-lactam antibiotics. There remains lack of clarity about broader-spectrum antibiotics and the authors continue to fail to define their variables. First, I will admit that until this version and response letter, I did not understand that the authors were only looking at broader-spectrum beta-lactam antibiotics. This remains unclear in the manuscript. The use of the abbreviation “BS-AB” is misleading; readers will naturally presume this refers to ALL broader spectrum antibiotics (not just beta-lactams). In the Methods, the authors say “there is no universally accepted definition for “broad-spectrum” antibiotics,” which is true. The rationale for looking only at broader-spectrum beta-lactam antibiotics is not explained until the Discussion section on page 13. This needs to be set up in the Introduction (why are they only looking at broader-spectrum beta-lactams and not a definition of all broader-spectrum antibiotics), explained in the Methods, and the Abstract and Table 1 need to be clarified. The use of the abbreviation “BS-AB” is misleading; readers will naturally presume this refers to ALL broader spectrum antibiotics (not just beta-lactam). In the Abstract Results – even though it is right there in the Abstract methods – the authors need to make it clear that “BS-AB” only refers to beta-lactams. For Table 1, the row headers at the bottom of alternate between specifying “beta-lactam” and not mentioning beta-lactams, which is misleading. In addition, the authors never define which beta-lactams are the “broader-spectrum beta-lactams.” Second, as currently written the Methods never defines or describes what is included in the “non-specific codes.” What about other, specified diagnoses not included in the 4 specific categories? Between the prior version and the present version, it is unclear what happened to the out-of-hours codes. The authors need to provide data to describe what is in this category, ideally with definitions and percentages in the Supplement or in a footnote to Table 2. The last sentence on page 9, paragraph 2 is inadequate. Third, the continued use of the “issueseq” terminology remains confusing. The manuscript never defines what this is or what is meant by “issueseq” or an “acute” prescription. Do the authors mean this is a “first” prescription? Presumably there are many chronic respiratory conditions (e.g., COPD) that require antibiotics. Instead of speaking in code-ese, the manuscript should define what the authors are seeking to describe. As I wrote before “The authors should define it in the Methods then use the actual, underlying concept throughout the rest of the manuscript.” Fourth, the revision does not include revised figures, so it is unclear how those might have changed. Specific Comments
--	---

	Page 6, Main Measures: this paragraph would be Page 8, 3rd paragraph: The sentence beginning “The proportion of registered patients...” is missing the word “each” before “year.” The sentence beginning “Figure 1 (left panel)...” should have the second instance of “in” replaced with “by.” Page 9, 1st paragraph: This paragraph seems out of place given that there is an entire subsection about changes in classes of antibiotics later on in the Results. Page 12, 1st paragraph: The newly-introduced sentence beginning with “However, these results suggested...” is out of place and disrupts the natural flow of ideas from the preceding and subsequent sentences. Page 14, 1st paragraph: The sentence beginning “Previous studies have demonstrated...” needs references.
--	---

VERSION 3 – AUTHOR RESPONSE

Reviewer: 1

I was very pleased to review this paper again. All of my concerns have now been addressed and I feel that this is a very interesting paper.

Thank you very much for this feedback and previous comments.

Reviewer: 2

-Thank you for the opportunity to review the second revision of the article by Sun and Gulliford, in their manuscript “Reducing antibiotic prescribing in primary care in England from 2014 to 2017: population-based cohort study.” I have modified my original comments below.

The investigators identified all antibacterial antibiotics prescribed in 102 English General Practices and measured changes in antibiotic prescribing between 2014 and 2017 per person-time stratified by age, gender, and indications, and whether the antibiotics were broader-spectrum beta-lactam antibiotics.

They found a 6.9% per year decrease in antibiotic prescribing, a 9.3% per year decline in broader-spectrum beta-lactam antibiotic prescribing, and noted a variety of differences by stratified variables.

Thank you for this feedback.

From the first revision to the present version the authors have addressed the representativeness of the data, provided a rationale for linear modeling of time, and addressed that they only looked at broader-spectrum beta-lactam antibiotics.

There remains lack of clarity about broader-spectrum antibiotics and the authors continue to fail to define their variables.

Thank you for helping us to enhance the quality of our paper. Each point has been addressed below.

First, I will admit that until this version and response letter, I did not understand that the authors were only looking at broader-spectrum beta-lactam antibiotics. This remains unclear in the manuscript. The use of the abbreviation “BS-AB” is misleading; readers will naturally presume this refers to ALL broader spectrum antibiotics (not just beta-lactams). In the Methods, the authors say “there is no universally accepted definition for “broad-spectrum” antibiotics,” which is true. The rationale for looking only at broader-spectrum beta-lactam antibiotics is not explained until the Discussion section on page 13. This needs to be set up in the Introduction (why are they only looking at broader-spectrum beta-lactams and not a definition of all broader-spectrum antibiotics), explained in the Methods, and the Abstract and Table 1 need to be clarified.

Thank you for your advice. We have now replaced the term ‘BS-AB’ with ‘broad-spectrum β -lactam antibiotic’ throughout. Specifically:

Abstract (page 2): this now refers to ‘broad-spectrum β -lactam antibiotics’.

Introduction (page 5): We now explain that ‘We compared changes in all antibiotic prescribing with changes in prescribing of broad-spectrum beta-lactam antibiotics.’

Methods: We now explain (page 6) that ‘We analysed as a separate group broad-spectrum β -lactam antibiotics including the British National Formulary category of ‘broad-spectrum penicillins’ (21) and cephalosporins. The category of ‘broad-spectrum penicillins’ includes ampicillin and amoxicillin and combinations with clavulanic acid or flucloxacillin. Carbapenems, which are only rarely used in primary care, were combined with cephalosporins for these analyses.’

Discussion: We now explain (page 13) ‘There is no universally accepted definition for ‘broad-spectrum’ antibiotics. This study analysed a separate category of β -lactam antibiotics that were broad-spectrum (as ‘broad-spectrum beta-lactam antibiotics’) to illustrate the possible difference in prescribing trends between these broad-spectrum antibiotics and their counterparts.’

The use of the abbreviation “BS-AB” is misleading; readers will naturally presume this refers to ALL broader spectrum antibiotics (not just beta-lactam). In the Abstract Results – even though it is right there in the Abstract methods – the authors need to make it clear that “BS-AB” only refers to beta-lactams. For Table 1, the row headers at the bottom of alternate between specifying “beta-lactam” and not mentioning beta-lactams, which is misleading.

Thank you for this suggestion. We now refer to ‘broad-spectrum beta-lactam antibiotics’ throughout the paper. The labelling of Table 2 has also been corrected.

In addition, the authors never define which beta-lactams are the “broader-spectrum beta-lactams.”

Thank you for this comment. As outlined above, we now explain (page 6) ‘We analysed as a separate group broad-spectrum β -lactam antibiotics including the British National Formulary category of ‘broad-spectrum penicillins’ and cephalosporins. The category of ‘broad-spectrum penicillins’ includes ampicillin and amoxicillin and combinations with clavulanic acid or flucloxacillin. Carbapenems, which are only rarely used in primary care, were combined with cephalosporins for these analyses.’

Second, as currently written the Methods never defines or describes what is included in the “non-specific codes.” What about other, specified diagnoses not included in the 4 specific categories? The authors need to provide data to describe what is in this category, ideally with definitions and percentages in the Supplement or in a footnote to Table 2. The last sentence on page 9, paragraph 2 is inadequate.

Thank you for this comment. We now provide additional explanation, together with a new Table 1 that shows the 30 most frequently employed codes in the category. We now explain (pages 6 to 7) 'All other codes were grouped into a single category of 'other and non-specific codes'. The most frequently used codes in this category are shown in Table 1 and included 'telephone encounter', 'patient reviewed' and 'telephone triage encounter'. Since specific coded indications for antibiotic therapy were infrequent in this category, it is subsequently referred to as 'non-specific' coding.

Between the prior version and the present version, it is unclear what happened to the out-of-hours codes.

Thank you, the category of 'out-of-hours' codes was omitted as a separate category because another Reviewer considered that it was not justified to separate this.

Third, the continued use of the "issuedseq" terminology remains confusing. The manuscript never defines what this is or what is meant by "issuedseq" or an "acute" prescription. Do the authors mean this is a "first" prescription? Presumably there are many chronic respiratory conditions (e.g., COPD) that require antibiotics. Instead of speaking in code-ese, the manuscript should define what the authors are seeking to describe. As I wrote before "The authors should define it in the Methods then use the actual, underlying concept throughout the rest of the manuscript."

Thank you for this comment. We now explain (page 7): 'We analysed the prescription sequence variable to determine whether each prescription was the first in a sequence or whether it was a repeat prescription; the former were coded as 'acute' prescriptions and the latter were coded as 'repeat' prescriptions.'

Fourth, the revision does not include revised figures, so it is unclear how those might have changed.

Thank you, the Figures were uploaded as separate pdf files and we will now check that these are incorporated into the pdf file.

Specific Comments

-Page 6, Main Measures: this paragraph would be

Thank you, this comment appears to be incomplete.

-Page 8, 3rd paragraph: The sentence beginning "The proportion of registered patients..." is missing the word "each" before "year." The sentence beginning "Figure 1 (left panel)..." should have the second instance of "in" replaced with "by."

Thank you. We have now corrected the text on page 8.

Page 9, 1st paragraph: This paragraph seems out of place given that there is an entire subsection about changes in classes of antibiotics later on in the Results.

Thank you for this comment, we wished to comment on these results in the order that they are presented. This paragraph refers to the data shown in Table 2.

-Page 12, 1st paragraph: The newly-introduced sentence beginning with "However, these results suggested..." is out of place and disrupts the natural flow of ideas from the preceding and subsequent sentences.

Thanks for this pertinent advice. Now we have reworded the text to elucidate the point in the first paragraph of page 12.

-Page 14, 1st paragraph: The sentence beginning “Previous studies have demonstrated...” needs references.

Thank you, we have now cited the appropriate reference.

VERSION 4 – REVIEW

REVIEWER	Jeffrey A. Linder, MD, MPH, FACP Division of General Internal Medicine and Geriatrics,,Northwestern University Feinberg School of Medicine, Chicago, IL USA
REVIEW RETURNED	21-Apr-2019

GENERAL COMMENTS	Summary Thank you for the opportunity to review the third revision of the article by Sun and Gulliford, in their manuscript “Reducing antibiotic prescribing in primary care in England from 2014 to 2017: population-based cohort study.” I have modified my original comments below. The investigators identified all antibacterial antibiotics prescribed in 102 English General Practices and measured changes in antibiotic prescribing between 2014 and 2017 per person-time stratified by age, gender, and indications, and whether the antibiotics were broader-spectrum beta-lactam antibiotics. They found a 6.9% per year decrease in antibiotic prescribing, a 9.3% per year decline in broader-spectrum beta-lactam antibiotic prescribing, and noted a variety of differences by stratified variables. The article now includes a clear description of the proportion of antibiotic prescriptions that are only associated with non-specific diagnosis codes (about 39% of all prescriptions) or no diagnosis codes (about 15% of all prescriptions). General Comments The topic is important and contains good and bad news. The good news is that there was a decrease in antibiotic prescribing in General Practice in England during the study. The decrease appears driven by decreases in the proportion of patients prescribed an antibiotic (or a broad-spectrum antibiotic) and by decreases in respiratory infection prescribing. The bad news is there remains a lot of antibiotic prescribing – potentially over half – that is at best poorly documented and at worst poorly indicated (other codes, out-of-hours prescribing, and no medical codes). The new version makes the rates of poor documentation and poorly indicated prescribing clear: it is about 54% of all antibiotic prescriptions!
--

	As I wrote before, the data are very interesting to those of us in antibiotic utilization research. From the initial submission to the first revision the investigators improved the discussion, aspects of the analysis (e.g., person-time, degree of prescribing captured), the structure of their multivariable models, the use (or non-use) of statistical testing, grammatical problems, indication coding, and implications of their findings. From the first revision to the second version, the authors addressed the representativeness of the data, provided a rationale for linear modeling of time, and addressed that they only looked at broader-spectrum beta-lactam antibiotics. From the second revision to the present version, the authors have clarified the rationale and description of the broader-spectrum beta-lactam antibiotic analysis; clarified the content of all of the diagnostic categories; and clarified descriptions of their variables. My only minor-to-moderate concern with the present version is that the abstract – and I recognize abstract word limitations makes this challenging – does not adequately present data that such a high proportion of antibiotic prescribing is poorly documented or has no indication. The Abstract Conclusion talks about the “Improving the quality of diagnostic coding for AB utilization...” but there are not data to support that assertion in the Abstract Results. A single sentence describing the top line results from Table 3 would suffice. This might require including an additional column in Table 3 combining all years and a corresponding sentence in the Results section of the methods. The authors are to be congratulated for putting a lot of work into this important analysis and manuscript.
--	--

VERSION 4 – AUTHOR RESPONSE

Response to comments:

Reviewer: 2

- General Comments

The topic is important and contains good and bad news. From the second revision to the present version, the authors have clarified the rationale and description of the broader-spectrum beta-lactam antibiotic analysis; clarified the content of all of the diagnostic categories; and clarified descriptions of their variables.

Thanks very much for these feedbacks as well as previous comments to help us improve our paper.

-My only minor-to-moderate concern with the present version is that the abstract. The Abstract Conclusion talks about the “Improving the quality of diagnostic coding for AB utilization...” but there are not data to support that assertion in the Abstract Results. A single sentence describing the top line results from Table 3 would suffice.

Thank you for your suggestion. We have now included the result about other and non-specific codes in accordance with our conclusion in the abstract, where it now reads: 'Overall, 38.8% of AB prescriptions were associated with codes that did not suggest specific clinical conditions and 15.3% of AB prescriptions had no medical codes recorded.'

This might require including an additional column in Table 3 combining all years and a corresponding sentence in the Results section of the methods.

A column that summarizes total AB prescriptions over study years has added up to Table 3 accompanying the ones for individual years.

The authors are to be congratulated for putting a lot of work into this important analysis and manuscript.

Thanks again for your help as always.